# Motivational Conflicts and the Psychological Structure of Perfectionism in Patients with Anxiety Disorders and Patients with Essential Hypertension

**DOI:** 10.3390/bs10010025

**Published:** 2020-01-01

**Authors:** Elena I. Pervichko, Yury A. Babaev, Anfisa K. Podstreshnaya, Yury P. Zinchenko

**Affiliations:** Faculty of Psychology, Lomonosov Moscow State University, 11/9 Mokhovaya St, Moscow 125009, Russia; yurybabaev@gmail.com (Y.A.B.);

**Keywords:** psychology, clinical, anxiety disorders, essential hypertension, motivation, perfectionism, projective techniques, achievement, power, psychological

## Abstract

Many studies have shown connections between perfectionism, motivation, and anxiety disorders (AD), as well as essential hypertension (EH). The objective of this study is to examine the connections between motivation and the structure of perfectionism in AD patients and EH patients compared to healthy individuals. Projective and semi-projective tests (thematic apperception test (TAT) of Heckhausen, Multi-Motive Grid) were used to measure motivation, while a perfectionism questionnaire by Hewitt and Flett was used to determine perfectionism levels. The participants were 21 AD patients, 21 EH patients, and 33 healthy individuals. EH patients show higher level of other-oriented perfectionism, and AD patients demonstrate increased levels of self-oriented perfectionism compared to the healthy group. Both groups of patients are motivated by fear of failure rather than hope for success, and they also demonstrate an increased fear of rejection. AD patients have an increased fear of power of other people. In EH patients, the fear of power seems to play a significant role as it correlates with many other variables. In EH patients, the other-oriented perfectionism is connected to achievement motivation, whereas in AD patients the same is true for socially prescribed perfectionism. Overall, studying motivation and perfectionism in relation to various disorders seems to offer research prospects.

## 1. Introduction

Anxiety disorders (AD) are referred to as some of the most widely spread mental disorders and present one of the major problems in psychiatry and clinical psychology today. According to different sources, 15%–33% percent of people experience AD during their lifetime [1,2]. Essential hypertension (EH) has in turn become a leading disease all over the world affecting younger populations [3]. Most authors agree on the significant influence of mental disorders, especially mood disorders, on the development and course of cardio-vascular diseases. Specifically, the interaction effect of the severity of depression and the triglyceride level was a modifiable factor positively associated with high cardiovascular risk [4]. Moreover, the connection between the two is reciprocal, i.e., patients with cardio-vascular diseases run a higher risk of developing mood disorders as well [5]. Because of their high comorbidity, some authors single out psychocardiology as a new subspecialty [6]. Etiology and pathogenesis of EH have not been researched in detail so far, though most researchers admit the important role of psychological factors in its initiation and development [7]. As personality traits are considered to be one of the essential groups of psychological factors affecting the origin and progress of EH [8], and the motivational sphere being the vital aspect in personality, it is considered important and practical to research the hierarchy of motives with EH patients. Not much is known about: the structure of the motivational sphere when afflicted with EH; its role in developing cognitive dysfunctions; its specific influence on choosing emotion-regulating strategies, unlike regular reciprocal influence between cognitive and emotional components of motivation which take place during activity; or the way they affect both physical and psychological health.

The motivational sphere is of major importance in the perspective of etiological treatment and prevention of the spectrum of AD. AD primarily affect the emotional component of psychological activity, and according to the approach of the theory of activity, emotions are treated as specific signals pointing to subjectively perceived successes or failures while fulfilling a certain action. Emotions reflect the relation between motives and the accomplishment of the activity corresponding to these motives [9]. Thus, emotional problems in the case of AD are supposed to be the evidence of motivational conflicts. Cognitive psychology and cognitive therapy deal with automatic thoughts, which are believed to determine the reactions of a subject to various life events (triggers) [10,11]. On a deeper level lie core beliefs, rules, and assumptions, which are clearly connected to human motivation (“I should do my best in every situation to be accepted and loved”, “I should control everything or something bad will happen” etc.). Motivation is also a central point in a psychodynamic approach, and motivational conflicts are believed to be connected to emotional disturbances [12,13]. Karen Horney pointed out that anxiety experienced by a person with neurosis is directly connected to perceived inability to satisfy important needs [14]. Rollo May expressed similar ideas in his works [15]. One may suggest that there should be considerable difference in motivation between patients with anxiety and the healthy population, however, there are few studies concerning motivation and AD. Researchers tend to view anxiety either from the neuropsychological point [16], or with application to academic motivation [17].

On the other hand, the link between perfectionism and anxiety is relatively well studied. Though perfectionism is primarily connected to depression, the latest research evidently shows that features of perfectionism are also typical for AD [18]. However, only a limited number of publications have analyzed perfectionism in reference to internal and external motivation. Similarly, some research has been done showing a connection between perfectionism and conflict between motivational tendencies of hope for success and fear of failure [19]. Some authors point out that the motivational basis of perfectionism has not been studied profoundly enough. The phenomenon of perfectionism is characterized by the presence of certain schemes and attitudes, through the prism of which an anxious person sees the world. The schemes of anxiety and harm avoidance were positively associated with depressive symptoms [20]. Nevertheless, the limitation of it to only a cognitive component seems to be an unreasonable simplification. It is evident that perfectionism has many aspects related to motivation [21]. They are dominant in the activity of failure avoidance motives combined with striving for abnormally high achievement, the urge to constantly assess others and compare themselves to others, the ambivalence of certain situations, etc. It is obvious that further research of perfectionism levels along with achievement-oriented motivations becomes immensely important.

The concept of Type A personality (linked to cardiovascular diseases by some authors) depicts some behavioral traits closely related to perfectionism: competitiveness, striving for high achievements, preference for difficult tasks, high responsibility, and deep involvement with work, etc [22]. These qualities were later related to perfectionism by a number of authors, despite being initially criticized for using parameters belonging to different areas, i.e. behavioral and motivational. The appropriateness of such an approach was disputed in further research and instead of behavioral type A, personality type D (distressed personality) has been widely discussed lately. Thus, the role of emotional stress and tension in cardiovascular diseases is emphasized. Type D describes a dismal, anxious kind of people avoiding social interaction due to potential risk of being criticized or underestimated [23]. From these descriptions, it is clear that emotional anxiety and EH are interconnected.

Perfectionism is a theoretical construct that is measured explicitly by questionnaires and represents conscious knowledge of the subject about their standards, primarily at the field of achievement, as well as attitudes toward errors in interpersonal context [24,25]. It is well known that explicitly and implicitly measured motives do not correlate [26], but any relation between motivation and perfectionism is rather indirect. For example, a positive answer to the statement, “Success means that I must work even harder to please others” does not imply high level of need for achievement, it may suggest a high level of fear of failure and disapproval of others. One can reasonably suggest that there might be some interaction between implicitly measured motivation and explicitly measured perfectionistic standards, and the kind of this interaction may also be specific to a certain disorder.

Several methodological approaches regard psychological disorders as interplay between some variables. For example, network analysis [27] suggests that for a specific disorder, a network of interacting symptoms can be constructed. These networks can provide scientifically and clinically useful information about core problems and help to set goals for psychotherapy. Such networks will also allow computer simulation for studying the impact of changing one of the variables (for example, anhedonia) on the others. One of the most influential Russian psychologists Lev Vygotsky proposed that there is a universal methodological approach to the study of psychological disturbances [28]. According to this approach, some disabilities (physical or psychological in nature) may result in the emergence of new qualities in personality during development, which serve as compensation for this defect. Although the design of the current research does not allow any conclusions to be drawn about the causal relationships between variables, one can suggest that explicitly measured variables (perfectionism and its subscales) may represent some sort of compensation for underlying “defects” in implicitly measured variables, i.e. motivation.

The main purpose of this study is to discover differences in motivation and perfectionism levels between healthy adults, patients with EH and patients with AD. Our hypothesis is that EH and anxiety patients will not only differ from healthy individuals (they will show higher perfectionism levels and they will be motivated primarily by fear, not hope), but will also reveal differing relationships between explicitly measured perfectionism and implicit motivation, that would imply compensatory relationships between those variables. Specifically, we expect to find statistically significant connections between self-oriented and other-oriented perfectionism and various components of motivation in EH patients and socially prescribed perfectionism and motivation in AD patients.

## 2. Materials and Methods

Two kinds of tests were used to measure motivation. The first of these is the projective thematic apperception test (TAT) developed by H. Heckhausen [29]. This test was developed as a variation of the classic TAT [30] for assessing achievement motivation. It measures two components of achievement motivation: “hope for success” (HS) and “fear of failure” (FF). In addition, two parameters are calculated based on HS and FF: “net hope” (NH = HS—FF) and “aggregate motivation” (AM = HS + FF). Second is Multi-Motive Grid by Schmalt, Sokolowski and Langens [31]. It was developed to assess motives in three domains: affiliation, achievement and power (or, in other words, motivation to gain control in social situations). For each group of motives, hope and fear components are measured, resulting in six variables: “hope for affiliation” (HA), “fear of rejection” (FR), “hope for success” (HS), “fear of failure” (FF), “hope for power” (HP) and “fear of power” (FP). This test is considered semi-projective, as participants are provided with possible answers for ambiguous picture stimuli. Both MMG and TAT measure hope for success and fear of failure components of achievement motivation, however, they appear to measure different psychological constructs as their results do not correlate [32].

The multidimensional perfectionism scale developed by Hewitt and Flett [33] is used for the assessment of perfectionism and its subscales. The scale measures three subscales of perfectionism: self-oriented (i.e. tendency to set high standards for oneself), other-oriented (high standards toward other people), and socially prescribed (the tendency of subjects to presume they have high standards imposed on them by the society).

The participants of the study were 21 treatment-naïve people with uncomplicated EH Stage 2 (mean age 49.3 ± 5.8), who were still not aware of their diagnosis at the time when the psychological tests were being taken. All EH patients underwent primary examination in cardiology department of the State Hospital #70, which is associated with Moscow State University of Medicine and Dentistry. They were included in the study only if their diagnosis of EH was later confirmed. The exclusion criteria were: heart attack or stroke, respiratory diseases, diseases of gastrointestinal tract or kidneys, hematological diseases and severe endocrine disorders, as well as diseases of the nervous system and history of mental disorders. 21 patients with AD were undergoing psychiatric treatment (F41.2 mixed anxiety and depressive disorder, with anxiety as leading syndrome, mean age 35.6 ± 10.6) in the National Medical Research Centre for Psychiatry and Narcology in Moscow. They were psychologically examined during their first days in clinic, before the onset of drug treatment. The exclusion criteria were: comorbid psychological disorders (such as personality disorder), EH or other cardiovascular disorders, severe respiratory, gastrointestinal, kidney, and endocrine or nervous system disorders. The control group consisted of 33 healthy individuals, who did not report history of mental or severe medical conditions (mean age 36.1 ± 8.3), and who were medically examined and did not show any symptoms of hypertension. The Hamilton anxiety scale [34] was used to test whether individuals in the healthy group demonstrated clinically significant levels of anxiety. Exclusion criteria for all participants were: age over 60 years, education level below secondary. All participants were volunteers and have given written informed consent prior to participation. Interuniversity Ethics Committee (Moscow, 119002, Gagarinsky per., 37) approved the study protocol (№ 04-18, 19.04.2018), all procedures performed in studies involving human participants were in accordance with the 1964 Helsinki Declaration and its later amendments.

After the test results were obtained, they were first evaluated for statistically significant differences between groups using one-way ANOVA. After that, the Pearson product-moment correlation test was used independently on all three groups to detect relationships between variables. All statistical tests were carried out with IBM SPSS Statistics software version 23.

## 3. Results

### 3.1. Differences between Groups

Differences in motivation and perfectionism levels were determined using the one-way ANOVA test. The Tukey HSD test was used for post hoc analysis. For self-oriented perfectionism, the total level of perfectionism and hope for achievement, as measured by the MMG test, the assumption of homogeneity of variances was not supported. For those variables, the Games-Howell test was used.

#### 3.1.1. Differences in Perfectionism Levels 

The mean and standard deviation values, as well as the results of one-way ANOVA for subscales and total perfectionism levels, are shown in Table 1.

Post hoc test procedures revealed the following differences: anxiety patients differ from healthy individuals (*p* = 0.010) and EH patients (*p* = 0.036) in self-oriented perfectionism. Patients with EH differ from healthy individuals (*p* = 0.016) and patients with AD in terms of other-oriented perfectionism. In addition, patients with an anxiety disorder show a statistically significant higher total level of perfectionism compared to the control group (*p* = 0.021).

Independent t-tests were run to clarify whether the difference existed between groups in socially prescribed perfectionism. It was found that anxiety patients differed from healthy individuals in socially prescribed perfectionism (t(52) = −2.181, *p* = 0.034). However, this result is not in concordance with ANOVA and should be interpreted with caution.

#### 3.1.2. Differences in Motivation

Results of one-way ANOVA for MMG test are shown in Table 2.

According to post-hoc tests, both anxiety patients (*p* = 0.025) and EH patients (*p* = 0.041) demonstrate higher rates of fear of rejection compared to the control group. In addition, patients with AD show increased fear of power compared to healthy individuals (*p* = 0.011).

The results of one-way ANOVA for Heckhausen TAT test are shown in Table 3.

Post hoc tests show a statistically significant difference in hope for success: in both anxiety patients (*p* = 0.000) and EH patients (*p* = 0.000) it is lower compared to the control group. EH patients show higher fear of failure than healthy individuals (*p* = 0.000) and patients with anxiety disorder (*p* = 0.006). Anxiety patients demonstrate lower overall level of achievement motivation than EH patients (*p* = 0.002) and healthy group (*p* = 0.000). In both EH (*p* = 0.000) and anxiety patients (*p* = 0.000) net hope is lower than in healthy individuals.

### 3.2. Correlations between Perfectionism and Motivation

A Pearson product-moment correlation was run to determine the relationship between different dimensions of perfectionism and motivation. All three groups were assessed separately.

For healthy individuals, there was only one statistically significant medium positive correlation between socially prescribed perfectionism and fear of failure measured by MMG (r = 0.356, *p* = 0.042).

In the group of EH patients, several statistically significant correlations were found between fear of power measured by MMG and other variables. These results are presented in Table 4.

Other-oriented perfectionism in EH patients demonstrated a medium negative correlation with fear of failure measured by TAT (r = −0.490, *p*= 0.024) and a rather strong positive correlation with net hope (r= 0.525, p = 0.014). Similar results were obtained for total perfectionism level, which correlates with fear of failure measured by TAT (r= −0.471, *p* = 0.031) and net hope (r= 0.442, *p* = 0.045). Self-oriented perfectionism in EH patients has medium negative correlation with hope for success measured by MMG (r = −0.482, *p* = 0.027).

In anxiety patients, socially prescribed perfectionism has strong negative correlation with hope for success measured by TAT (r = −0.533, *p* = 0.013). Total level of perfectionism correlates negatively with hope for affiliation measured by MMG (r= −0.435, *p* = 0.049). In addition, hope for affiliation measured by semi-projective MMG correlates with net hope, measured by TAT (r = 0.434, *p* = 0.049).

## 4. Discussion

The results of the study suggest that there are considerable differences in perfectionism levels between groups. EH patients tend to set higher standards for others, which supports evidence for their other-directedness and sometimes aggressive demands toward other people. Similar conclusions were made by Flett et al. when studying perfectionism and its connection to type A behavior. The authors show that interpersonal conflicts and hostility in relationships might be provoked due to unrealistic standards towards others set by people with Type A behavior [35]. Anxiety patients in turn are more self-oriented in terms of perfectionism and prone to setting unrealistically high standards for themselves. This is not supported by previous research, as self-oriented perfectionism is supposed to be related to depression, and socially prescribed perfectionism is connected to anxiety [18,36]. This is probably explained by the fact that the anxiety group included patients with mixed diagnosis.

We expected to find statistically significant differences between the two groups of patients and the control group, since recent research connects socially prescribed perfectionism with psychopathology [18,37]. The direct comparison of the anxiety group with healthy individuals confirms this hypothesis, but this evidence is not supported by the ANOVA test. New studies with bigger sample sizes may be needed to confirm the existence of this relationship.

In terms of motivation, both groups of patients show higher rates of fear of rejection than the healthy group. In addition, anxiety patients demonstrate an increased fear of the power of others. 

Fear of power within EH patients seems to be a very important parameter because of its significant association with many domains. It has an influence on a wide variety of social aspects and might explain the fact that EH patients tend to overreact in social situations where there is a perceived threat of losing control. Previous research confirmed the association between the increase of blood pressure and reaction to experimental arousal of the power motive and social dominance provocation in people who are driven primarily by the motivation of power [38]. Striving for affiliation in EH patients is expressed by the dominance of fear of being rejected. This constant tension in social situations might result in chronic psychosocial stress, which is considered to play a significant role in the development of EH and is emphasized by many authors [39]. Elevated levels of other-oriented perfectionism and demands toward others might represent a compensatory mechanism for underlying fear of power and losing control in EH patients.

The analysis of achievement motivation measured implicitly by TAT resulted in the highest effect sizes. Both groups of patients show lower level of hope for success than control group. Moreover, their motivation is guided primarily by fear of failure and not by hope for success. Fear of failure is higher in EH patients compared to the other two groups. Patients with anxiety demonstrate lower overall level of achievement motivation than EH patients and healthy individuals.

Both groups of patients are other-directed in terms of perfectionism and motivation, but types of this “other-directedness” seem to be different in quality. EH patients tend to compensate for their fear of power of others using high demands towards those around them. Patients with anxiety, in turn, tend to set high standards for themselves in order to be “good” and accepted by others. In both groups, the fear of failing and punishment from others prevails over hope for success. As both EH patients and patients with anxiety disorder may appear as overachievers, they might do it not in order to succeed, but in their constant striving not to fail.

### 4.1. Strengths of the Study

Theoretical background of the study allows integrating clinical psychology approaches used in Russia and other countries, the results of the study correspond to the theoretical background, which clearly presents a perspective for future research. The groups of patients were thoroughly examined by clinicians to exclude comorbidity. Involving treatment-naïve voluntary patients allowed excluding factor of disease awareness.

### 4.2. Limitations of the Study

The reliability of the study could be improved by follow-up studies and the increasing of sample sizes. The correlational nature of the study does not allow for any conclusions about causal relationships of explicit and implicit factors to be drawn. Additional studies concerning different types of AD should be carried out to clarify relationships between perfectionism and motivation in patients with anxiety.

## 5. Conclusions

Overall, complex relationships seem to exist between motivation and perfectionism for both essential hypertension and anxiety groups. This is not true for healthy individuals, so one may suggest that these relationships are related to psychological causes of the disorder. These findings may be used in clinical practice to increase compliance, set goals for psychotherapy and strengthen therapeutic alliance. This is of particular importance in the case of anxiety disorders, as dropout rates among those patients are considerable [40]. For example, in patients with anxiety, the combination of high personal standards and fear of failure may result in dropout, because of the patient’s fear of not being “a good enough patient” and fear of rejection by the authoritative figure of their therapist. These high standards and orientation towards failure avoidance may serve as goals for psychotherapy from the early stages onwards. EH patients in contrast may reject their clinician because of having high demands towards him/her and underlying competition for control over the process of therapy. In the future, psychological intervention is required to address this competition of control as it may affect adherence to treatment. Previous research found that cardiac patients who had poor adherence to cardiac rehabilitation reported lower levels of physical and mental quality of life and higher levels of depression post-cardiac rehabilitation [41].

The approach that views a psychological disorder as an interplay of explicit and implicit factors seems to have potential. Future research may consider depression and its comparison to anxiety disorders in terms of perfectionism and motivation, as well as distinguishing between different anxiety disorders, post-traumatic stress disorder, etc. The relationship between somatic disorders such as essential hypertension and underlying psychological factors also remains among the most important problems of clinical psychology.

## Figures and Tables

**Table 1 behavsci-10-00025-t001:** Differences in levels of perfectionism.

Variable	Healthy		EH Patients		AD Patients		F(2,72)	Sig.
	M	SD	M	SD	M	SD		
Self-oriented ^1^	57.24	17.12	56.52	19.21	69.43	12.19	4.30	0.017 *
Other-oriented	56.42	13.57	68.14	18.73	56.43	12.00	4.73	0.012 *
Socially prescribed	56.76	10.55	62.10	9.93	63.62	12.34	2.99	0.057
Total level ^1^	170.42	24.74	186.76	39.47	189.48	24.37	3.37	0.040 *

Note: ^1^ Homogeneity of variances was violated, * *p* < 0.05.

**Table 2 behavsci-10-00025-t002:** Differences in motivation measured by MMG.

Variable	Healthy		EH Patients		AD Patients		F(2,72)	Sig.
	M	SD	M	SD	M	SD		
HA ^1^	7.52	2.83	7.57	1.69	7.62	1.53	0.01	0.986
FR	4.94	2.86	6.90	2.23	6.76	2.62	4.79	0.011 *
HS	8.09	2.49	7.10	2.30	7.57	2.54	1.08	0.346
FF	5.39	2.84	6.05	2.33	6.76	2.47	1.79	0.175
HP	7.79	2.89	8.19	2.82	7.10	3.22	0.74	0.482
FP	5.48	2.67	6.90	2.98	7.62	1.86	4.83	0.011 *

Note: ^1^ Homogeneity of variances was violated, * *p* < 0.05.

**Table 3 behavsci-10-00025-t003:** Differences in motivation measured by Heckhausen TAT.

Variable	Healthy		EH Patients		AD Patients		F(2,72)	Sig.
	M	SD	M	SD	M	SD		
Hope for success	11.73	4.35	6.14	2.59	4.81	3.46	27.203	0.000 **
Fear of failure	6.09	3.21	10.90	3.13	7.71	3.36	14.266	0.000 **
Aggregate	17.82	4.95	17.05	3.88	12.52	4.08	9.762	0.000 **
Net hope	5.64	5.84	−4.76	4.24	−2.90	5.47	29.778	0.000 **

Note: ** *p* < 0.01.

**Table 4 behavsci-10-00025-t004:** Correlations between fear of power and other variables in EH patients.

Variable	R	Sig.
Fear of rejection (MMG)	0.629	0.002 **
Fear of failure (MMG)	0.532	0.013 *
Soc. prescribed perfectionism	−0.516	0.017 *
Total perfectionism	−0.449	0.041 *
Hope for success (TAT)	−0.464	0.034 *
Net Hope (TAT)	−0.583	0.006 **

Note: * *p* < 0.05, ** *p* < 0.01.

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
