# Peer review of "Motivational Conflicts and the Psychological Structure of Perfectionism in Patients with Anxiety Disorders and Patients with Essential Hypertension"

_behavsci, 2020, doi:10.3390/bs10010025_

Round 1

Reviewer 1 Report

Thank you for inviting me to review the paper on “Motivational Conflicts and the Psychological Structure of Perfectionism in Patients with Anxiety Disorders and Patients with Essential Hypertension”. This is a good study with sound methodology and deserves to be published. I have the following recommendations and happy to review the paper again.

Under the introduction, Line 32-33 the authors stated, “Most authors agree on the significant influence of mental disorders, especially mood disorders, on the development and course of cardio-vascular diseases.” This statement is vague, without details and lack of references. I suggest to add the following statement:

.. s, on the development and course of cardio-vascular diseases. Specifically, the interaction effect of the severity of depression and the triglyceride level was a modifiable factor positively associated with high cardiovascular risk (Ho et al 2018).

Reference:

Factors Associated with the Risk of Developing Coronary Artery Disease in Medicated Patients with Major Depressive Disorder. Ho RCM, et al. Int J Environ Res Public Health 2018 PMID 30248896 

Under the Introduction, Line 70, the authors stated that “The phenomenon of perfectionism is characterized by the presence of certain schemes and attitudes through the prism of which an anxious person sees the world.” It is important to highlight how certain schemes are associated with depression. Please add the following statement:

… anxious person sees the world. The schemes of anxiety and harm avoidance were positively associated with depressive symptoms (Lim et al 2018).

Reference:

Lim CR et al. The Effects of Temperament on Depression According to the Schema Model: A Scoping Review. Int J Environ Res Public Health. 2018;15(6):1231. doi:10.3390/ijerph15061231. PMID: 29891793

Under methodology, line 136, the authors need to clarify the meaning of “naïve” people.

Under discussion, line 255, the authors stated that “EH patients in contrast may reject their clinician because of having high demands towards him/her and underlying competition for control over the process of therapy.” It is important to elaboration on the clinical application of this finding. Please add the following statement:

….. In the future, psychological intervention is required to address this competition of control as it may affect adherence to treatment. Previous research found that cardiac patients who had poor adherence to cardiac rehabilitation reported lower levels of physical and mental quality of life and higher levels of depression post-cardiac rehabilitation (Choo et al 2018).

Reference

Effect of Cardiac Rehabilitation on Quality of Life, Depression and Anxiety in Asian Patients.

Choo CC, et al. Int J Environ Res Public Health 2018 PMID 29843421

Author Response

We would like to thank you for your time and effort invested in the review and for your valuable suggestions. You may find corrected manuscript in attachment. Here is what we have done to improve our article according to your comments:

Under the introduction, Line 32-33 the authors stated, “Most authors agree on the significant influence of mental disorders, especially mood disorders, on the development and course of cardio-vascular diseases.” This statement is vague, without details and lack of references. I suggest to add the following statement:
.. s, on the development and course of cardio-vascular diseases. Specifically, the interaction effect of the severity of depression and the triglyceride level was a modifiable factor positively associated with high cardiovascular risk (Ho et al 2018).

We have updated the text and references.

Under the Introduction, Line 70, the authors stated that “The phenomenon of perfectionism is characterized by the presence of certain schemes and attitudes through the prism of which an anxious person sees the world.” It is important to highlight how certain schemes are associated with depression. Please add the following statement: anxious person sees the world. The schemes of anxiety and harm avoidance were positively associated with depressive symptoms (Lim et al 2018).

We have updated the text and references.

Under methodology, line 136, the authors need to clarify the meaning of “naïve” people.
We changed "naïve" to "treatment-naïve" which is more commonly used and better reflects the subject.

Under discussion, line 255, the authors stated that “EH patients in contrast may reject their clinician because of having high demands towards him/her and underlying competition for control over the process of therapy.” It is important to elaboration on the clinical application of this finding. Please add the following statement: In the future, psychological intervention is required to address this competition of control as it may affect adherence to treatment. Previous research found that cardiac patients who had poor adherence to cardiac rehabilitation reported lower levels of physical and mental quality of life and higher levels of depression post-cardiac rehabilitation (Choo et al 2018).

We have updated the text and references.

Reviewer 2 Report

The paper entitled “Motivational conflicts and the psychological structure of perfectionisms in patients with anxiety disorders and patients with essential hypertension” by Pervichko et al. consists in an investigation of the connections between motivation and the structure of perfectionism in patients with anxiety disorders (AD) and essential hypertension (EH) compared to healthy individuals through the use of projective and semi-projective tests (TAT, Multi-Motive Grid), and a perfectionism questionnaire. The aim of the study is interesting and the methodology is well described. However, some points have to be revised, as suggested:

I suggest the authors to revise the key words considering the suggestions of MeSH terms.

Endpoints of the study should be clarified.

In Methods section, the part on recruitment of participants and selection criteria is missing and it has been reported only in the abstract. Moreover, I suggest the author to report the inclusion and exclusion criteria used.

I suggest the authors to report the characteristics of the setting and the eventual approval of the study by the Ethics committee with related protocol number.

Finally, I suggest the authors to report strength and limitations of the study.

In light of these considerations, the paper could be valuable for publication after revisions.

Author Response

We would like to thank you for your time and effort invested in the review and for your valuable suggestions. You may find corrected manuscript in attachment. Here is what we have done to improve our article according to your comments:

I suggest the authors to revise the key words considering the suggestions of MeSH terms.

We have changed the key words so that they represent preffered MeSH terms.

Endpoints of the study should be clarified.

We reformulated the last paragraph of the introduction to clarify endpoints of the study (lines 116-125).

In Methods section, the part on recruitment of participants and selection criteria is missing and it has been reported only in the abstract. Moreover, I suggest the author to report the inclusion and exclusion criteria used.

I suggest the authors to report the characteristics of the setting and the eventual approval of the study by the Ethics committee with related protocol number.

We have updated the Methods section to clarify the mentioned issues

Finally, I suggest the authors to report strength and limitations of the study.

Strength and limitations of the study were added at the end of the "Discussion" section

Round 2

Reviewer 1 Report

I recommend publications.

Reviewer 2 Report

Authors have followed the reviewer suggestions.